# Transfer learning in Scalable Graph Neural Network for Improved Physical Simulation

## Abstract

In recent years, Graph Neural Network (GNN) based models have shown promising results in simulating physics of complex systems. However, training dedicated graph network based physics simulators can be costly, as most models are confined to fully supervised training, which requires extensive data generated from traditional physics simulators. To date, how transfer learning could improve the model performance and training efficiency has remained unexplored. In this work, we introduce a pre-training and transfer learning paradigm for graph network simulators. We propose the scalable graph U-net (SGUNet). Incorporating an innovative depth-first search (DFS) pooling, the SGUNet is adaptable to different mesh sizes and resolutions for various simulation tasks. To enable the transfer learning between differently configured SGUNets, we propose a set of mapping functions to align the parameters between the pre-trained model and the target model. An extra normalization term is also added into the loss to constrain the difference between the pre-trained weights and target model weights for better generalization performance. To pre-train our physics simulator we created a dataset which includes 20,000 physical simulations of randomly selected 3D shapes from the open source A Big CAD (ABC) dataset. We show that our proposed transfer learning methods allow the model to perform even better when fine-tuned with small amounts of training data than when it is trained from scratch with full extensive dataset. On the 2D Deformable Plate benchmark dataset, our pre-trained model fine-tuned on 1/16 of the training data achieved an 11.05% improvement in position RMSE compared to the model trained from scratch.

## 1 Introduction

Graph Neural Networks (GNNs) have shown promising results in simulating physics of complex systems on unstructured meshes Pfaff et al. (2021); Allen et al. (2022a); Rubanova et al. (2022); Allen et al. (2022c). Existing works stack message passing (MP) blocks to model propagation of physical information. Different pooling operations Li et al. (2020); Cao et al. (2023) and U-Net like architectures Gladstone et al. (2023); Deshpande et al. (2024) have been introduced to better solve the multi-scale challenges in different simulation tasks. However, despite their potential, current GNN-based methods rely heavily on supervised training approaches. Collecting extensive annotated data typically involves using traditional Finite Element Analysis (FEA) solvers Han et al. (2022); Fortunato et al. (2022); Allen et al. (2022b), 3D engines Greff et al. (2022), and real-life video clips Lopez-Guevara et al. (2024). The substantial cost of acquiring the training data has constrained the scalability and practicality of GNN-based simulators.

On the other hand, transfer learning has revolutionized fields like computer vision (CV) Reddy & Juliet (2019); Rezende et al. (2017) and natural language processing (NLP) Radford et al. (2019); Mann et al. (2020); Touvron et al. (2023), where models pre-trained on large datasets are fine-tuned for specific tasks, leading to remarkable improvements in training efficiency and the model's performance Zhuang et al. (2020). However, for GNN-based simulators, network architecture hyperparameters, such as number of message passing steps and pooling ratios, must be specifically tailored to the mesh resolution of the target problem Fortunato et al. (2022); Cao et al. (2023). As a result, pre-trained models are difficult to load and fine-tune directly for downstream GNN-based simulators. It remains unexplored how to apply transfer learning to GNN-based physics simulators.

In this work, we introduce a transfer learning paradigm applied to the proposed scalable graph U-net (SGUNET) for physical simulations. The SGUNET follows the Encoder-Processor-Decoder design and incorporates an innovative DFS pooling operation to handle various mesh receptive fields. It is designed to be modular and configurable, making it adaptable to different mesh sizes and resolutions across various simulation tasks. To enable transfer learning between differently configured SGUNETs, we propose a set of mapping functions to align the parameters between the pre-trained model and the target model. An extra normalization term is also added into the loss to constrain the difference between the pre-trained weights and target model weights for better generalization performance. As there is no existing dataset available for pre-training, we created a dataset named ABC Deformable (ABCD) for pre-training. The dataset includes approximately 20,000 physical simulations of deformable bodies, whose shapes are sampled from the open-source ABC dataset Koch et al. (2019).

We evaluated our proposed methods over two public datasets, namely the 2D Deformable Plate Linkerhägner et al. (2023) and a more complex 3D Deforming Plate Pfaff et al. (2021). We set MESHGRAPHNET (MGN) Pfaff et al. (2021) training from scratch as the baseline. On 2D Deformable Plate, our model pre-trained by ABCD and fine-tuned on 1/16 of the training data could achieve an 11.05% improvement in position RMSE compared to the model trained from scratch. On 3D Deforming Plate, our pre-trained model reached the same level of performance when fine-tuned with only 1/8 of the training data in 40% of the training time. Applying the transfer learning approach to MGN also lead to better performance with less training data and shorter training time.

## 2 RELATED WORK

Graph Neural Networks (GNNs) have emerged as a powerful tool for simulating complex physical systems, particularly on unstructured meshes Pfaff et al. (2021); Allen et al. (2022a); Rubanova et al. (2022); Allen et al. (2022c;b). However, these methods predominantly rely on supervised training, which requires extensive annotated data. Common approaches involve generating data through analytical solvers like OpenFOAM Weller et al. (1998) and ArcSim Narain et al. (2012). Additionally, some works use real-world observations to train models Whitney et al. (2023); Allen et al. (2022c). Early work, such as MGN Pfaff et al. (2021), adapts the Encoder-Process-Decode architecture Sanchez-Gonzalez et al. (2020) to mesh data, with the Process module implemented as a GNN for effective message passing. Variants like EA-GNN and M-GNN Gladstone et al. (2023) introduce enhancements such as virtual edges and multi-resolution graphs to improve efficiency and handle long-range interactions. Additionally, the transformer architecture has been explored in mesh-based physical simulations. Hybrid models like the GMR-Transformer-GMUS Han et al. (2022) and HCMT Yu et al. (2023) combine GNNs to learn local rules and transformers to capture global context and long-term dependencies over roll-out trajectories. Unlike most methods that directly predict future states from input data, C-GNS Rubanova et al. (2022) employs a GNN to model system constraints and computes future states by solving an optimization problem based on these learned constraints.

Transfer learning, which transfers knowledge from a source domain to a target domain, has gained prominence in deep learning for improving performance and reducing the need for annotated data Reddy & Juliet (2019); Rezende et al. (2017); Mann et al. (2020); Touvron et al. (2023). Strategies typically involve parameter control, either by sharing parameters between models or enforcing similarity through techniques like $l^2$-norm penalties Zhuang et al. (2020); Gouk et al. (2020); Xuhong et al. (2018). These approaches have proven effective in computer vision and natural language processing. However, the application of transfer learning to GNN-based physics simulations remains largely unexplored.

## 3 METHOD

### 3.1 OVERVIEW

In this section, we introduce our pre-training and fine-tuning framework. We begin by detailing the data format used by our model. Following this, we provide an in-depth explanation of the model architecture, discussing its fundamental networks, operators, and key modules. Finally, we describe

the transfer learning mechanism, emphasizing two mapping functions that adjust the model size for optimal performance.

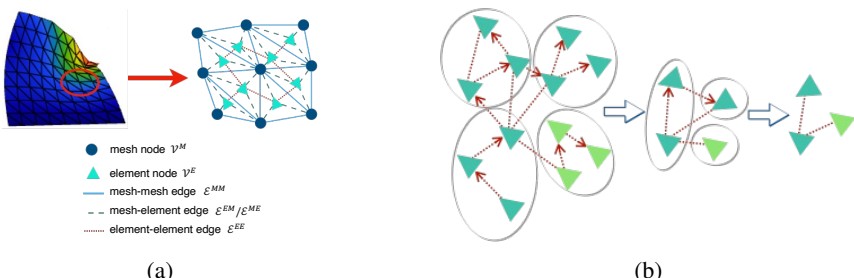

Figure 1: (a) An illustration of the composition of mesh data $\mathcal{M}$ and its representation as a heterogeneous graph $\mathcal{G}^{hetero}$. (b) Example of down-sampled graphs with pooling ratios as 3 and 2.

### 3.2 PROBLEM STATEMENT

Given the physical state of a system, our task is to predict the subsequent state over a time interval and under specified boundary conditions. The system's current state at time $t$ is described by a discretized mesh $M^t$ and can be represented in 2D or 3D space. The mesh data $M^t$ is comprised of world coordinates, element connectivity, and physical parameters (stress/strain state and material properties). We use one-step prediction to find the subsequent mesh state $M^{t+1}$, but for sake of notational convenience, we will omit the superscript $t$, which indicates the timestamp in the subsequent expressions.

To facilitate the learning process, we transform the original mesh data $M$ into a heterogeneous graph $G^{hetero} = (\mathcal{V}, \mathcal{E})$, where $\mathcal{V}$ denotes the set of nodes and $\mathcal{E}$ represents the set of edges. The node set $\mathcal{V}$ comprises two types of node: mesh nodes $\mathcal{V}^M$ and element nodes $\mathcal{V}^E$. The mesh vertices in $M$ are converted to graph nodes in $\mathcal{V}^M$, while the mesh faces are represented as nodes in $\mathcal{V}^E$. This heterogeneous graph structure is necessary to describe physical state variables like stress in a single-valued way when multiple materials are present in a system, as is typical for many physical simulations. The edge set $\mathcal{E}$ includes three groups of edges: (1) bidirectional edges $\mathcal{E}^{M,M}$ between adjacent mesh vertices $\mathcal{V}^M$, (2) bidirectional edges $\mathcal{E}^{E,E}$ between adjacent faces $\mathcal{V}^E$, and (3) directional edges $\mathcal{E}^{E,M}$ and $\mathcal{E}^{M,E}$ which connect each mesh face to its vertices. Figure 1a provides a demonstration of mesh data derived from the Deformable Plate dataset and its corresponding heterogeneous graph.

Each type of node and edge has its own feature matrix. For example, the feature matrix of mesh nodes $\mathcal{V}^M$ is $\mathbf{X}^M \in \mathbb{R}^{|\mathcal{V}^M| \times h_0^M}$, where the feature vector of node $i \in \mathcal{V}^M$ is the $i$-th row vector of $\mathbf{X}^M$. We explain the composition of feature matrices in Appendix **??**.

### 3.3 SCALABLE GRAPH U-NET

Our model comprises four main modules, as illustrated in Figure 2a. We adopt an Encoder-Process-Decoder Sanchez-Gonzalez et al. (2020) style model with several key extensions: (1) the framework is extended to handle the heterogeneous graph structure with multiple node and edge types, and (2) we add a staged U-Net with variable DFS pooling and unpooling that greatly expands the model's receptive field. In the following paragraphs, we will detail these networks and modules.

**Encoder:** The Encoder comprises a set of MLP models and three Processors. The MLP models map raw node and edge features of varying sizes into a unified latent space. The three Processors manage the message passing flows $\mathcal{V}^M \to \mathcal{V}^M$, $\mathcal{V}^E \to \mathcal{V}^E$, and $\mathcal{V}^M \to \mathcal{V}^E$, respectively. Consequently, the Encoder transforms the heterogeneous graph $\mathcal{G}^{hetero}$ into a homogeneous graph $\mathcal{G}^E$ by aggregating the mesh node representations into their neighboring element nodes.

**Processor:** The Processor (Pr) consists of $m$ identical Graph-Net blocks (GNBs), with the output of each block serving as the input to the subsequent block in the sequence. This sequential processing enables the model to gather and integrate information from nodes located up to $m$ hops away from the central node.

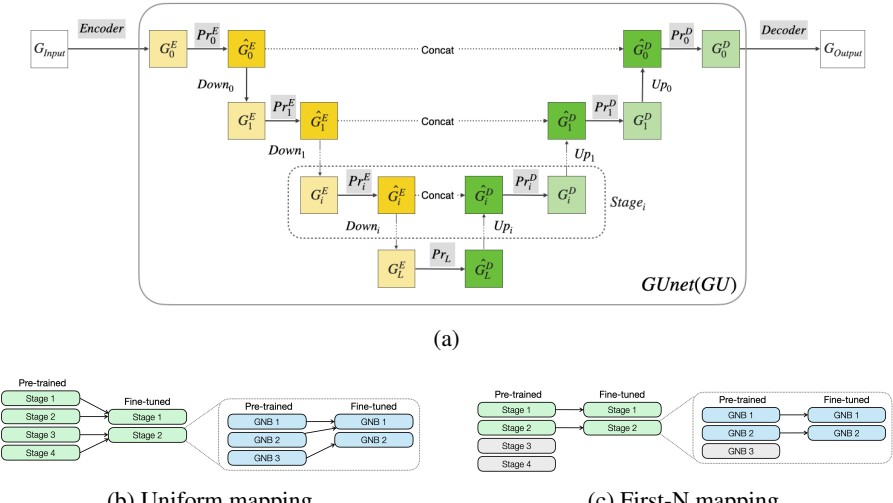

(a)

(b) Uniform mapping           (c) First-N mapping

Figure 2: (a) A detailed depiction of our proposed model, SGUNET, which includes four primary modules: the **Processor** $Pr_i$ for information propagation; the **Encoder** for data transformation, the **GUnet** for graph pooling, and the **Decoder** for downstream tasks. (b) & (c) Mapping functions for GUNet stages and GNBs for the case where pre-trained model has more stages and per-processor GNBs than the fine-tuned model.

Each Graph-Net block operates as an independent message passing unit, with no shared parameters between blocks. We extend the concept of Graph-Net blocks from Sanchez-Gonzalez et al. (2018); Pfaff et al. (2021) to accommodate graphs with various types of nodes and edges. Specifically, for message passing flow from $\mathcal{V}^{\text{src}}$ to $\mathcal{V}^{\text{tgt}}$, where src and tgt $\in \{E, M\}$, the feature of the edge connecting nodes $i$ and $j$ is updated via

$$\mathbf{X}_{i,j}^{\text{src,tgt}} = f^{\text{src,tgt}} \left( \mathbf{X}_i^{\text{src}} | \mathbf{X}_j^{\text{tgt}} | \mathbf{X}_{i,j}^{\text{src,tgt}} \right),$$

where $f^{\text{src,tgt}}$ is an MLP model with residual connections, and $|$ represents the concatenation operation. After updating the edge features, the feature of the end-point node $j$ is updated via

$$\mathbf{X}_j^{\text{tgt}} = f^{\text{tgt}} \left( \mathbf{X}_j^{\text{tgt}} | \sum_{i \in \mathcal{N}_j^{\text{src}}} \mathbf{X}_{i,j}^{\text{src,tgt}} \right),$$

where $f^{\text{tgt}}$ is an MLP model with residual connections, and $\mathcal{N}_j^{\text{src}}$ are the neighborhoods of node $j$ of type src.

**GUnet Stage:** The GUnet (GU) is composed of $L$ stages, with each stage containing one pooling layer and two Processors. During the down-sampling phase, a Processor ($\text{Pr}_i^E$) and a down-sample operator ($\text{Down}_i$) are sequentially applied at stage $i$. This process allows graphs of varying sizes to undergo information propagation and aggregation. During the up-sampling phase, the procedure is reversed: the graph is first restored to its original size by an up-sampling operator ($\text{Up}_i$) before being processed by a Processor ($\text{Pr}_i^D$) at stage $i$. Unlike traditional graph network models such as those in Pfaff et al. (2021); Rubanova et al. (2022); Allen et al. (2022a), which require many message passing steps to capture long-range dependencies, the GUnet is able to process long-range data with significantly fewer message passing steps. This can also help alleviate the difficulty of scaling message passing networks to a large number of nodes as the number of message passing steps needed would become intractable for GPU memory.

We design a Depth First Search style pooling (DFS-pooling) which has two key advantages that distinguishes it from the existing work Gao & Ji (2019): (1) changeable pooling ratios for different pooling stages and (2) pooling based on node proximity. We first select an un-visited node from the element node set as the starting node and initiate a DFS-style random walk to explore adjacent nodes of the same material. During this walk, nodes are clustered according to the pooling ratio, with nodes in each cluster pooled into a single node. We use an even-pooling scheme of averaging weights to

update the node and edge features within each cluster. This process continues until all nodes in $\mathcal{V}^E$ have been visited. The pseudo-code for this clustering computation is provided in Algorithm 1. Additionally, we provide an example of down-sampled graphs in Figure 1b with pooling ratios as 3 and 2, respectively. Here, nodes of different colors have different types of material. Note that this computation needs to be performed only once during the preprocessing stage, once the pooling ratio is determined.

**Decoder:** In line with MGN Pfaff et al. (2021), we use an MLP model to project the latent features of the element nodes $\mathbf{X}^E$ into the output space. Additionally, we perform a 1-hop message passing operation from $\mathcal{V}^E$ to $\mathcal{V}^M$ to interpolate the features of the element nodes to those of mesh nodes. Finally, another MLP model decodes the interpolated latent features of the mesh nodes into the desired output space.

### 3.4 UTILIZING THE PRE-TRAINED MODEL

How exactly the knowledge from a pre-trained model is instilled into a target task-specific model, particularly when their architectures are mismatched, is a non-trivial task. As stated in Section 2, one of the widely-used approaches in transfer learning operates at the parameter level. We also adopt the parameter sharing and parameter restriction strategies and discuss these in more detail here. To our knowledge, this is the first time transfer learning has been adapted and applied to GNNs predicting physics simulations.

#### 3.4.1 PARAMETER SHARING

The parameter sharing strategy involves initializing the downstream task-specific model with a pre-trained model. Typically, the pre-trained model is either the same as or more complex than the task-specific model due to resource constraints Xu et al. (2023). However, this is not the case for our mesh-based graph network model. This distinction arises because the number of stages and message passing steps in the GUnet are closely aligned with the data size and simulation settings. As a result, conventional weight initialization methods are not directly applicable. Therefore, we design alternative strategies to effectively transfer learned knowledge from the pre-trained model to the downstream model within our unique framework.

We propose a scaling method at two levels — Processor and GUnet — to align the sizes of the pre-trained and fine-tuned models. We choose between two mapping functions for both the Processors and GUnet: Uniform and First-N. The details of these mapping functions are described below. For the Processors, we employ the mapping function on GNBs, which ensures that the alignment reflects the stages of message propagation through the blocks. For the GUnet, we employ the mapping function on the GUNet stages. In general, the mapping function chosen for the Processors and GUnet need not be the same, but for our present work we used the same mapping (First-N or Uniform) for both Processors and GUnet for a given experiment.

Note that the Encoder and the Decoder do not contain generalizable knowledge as they are tailored to specific tasks. In our implementation, we randomly initialize the parameters of the Encoder and Decoders during fine-tuning.

**1) Uniform Mapping:** The first method of parameter sharing between pre-trained and fine-tuned models is the Uniform method. The goal of this mapping method is to achieve uniform division and alignment of weights. To align the weights of two Processors, we consider cases where the number of GNBs in the pre-trained model $m_{\text{pt}}$ is less than, greater than, or equal to that in the fine-tuned model $m_{\text{ft}}$. Formally, let $g_1(\mathbf{Pr}_{\text{pt}})$ be a function that maps a Processor $\mathbf{Pr}_{\text{pt}}$ in the pre-trained model to a Processor $\mathbf{Pr}_{\text{ft}}$ in the fine-tuned model. This mapping function uses uniform division to achieve the alignment:

$$\mathbf{Pr}_{\text{ft}}^i = g_1^{\text{uni}}(i; \mathbf{Pr}_{\text{pt}}) = \begin{cases} \mathbf{Pr}_{\text{pt}}^{\lfloor i/\text{upN} \rfloor}, & \text{if } m_{\text{pt}} < m_{\text{ft}} \\ \text{MEAN}_{j=st(i)}^{st(i)+d(i)}\{\mathbf{Pr}_{\text{pt}}^j\}, & \text{if } m_{\text{pt}} > m_{\text{ft}} \\ \mathbf{Pr}_{\text{pt}}^i, & \text{if } m_{\text{pt}} == m_{\text{ft}}, \end{cases}$$

where,

$$st(i) = \begin{cases} (\text{dwN} + 1) \times i, & \text{if } i < \text{r}, \\ \text{dwN} \times i + \text{r}, & \text{if } i \geq \text{r}, \end{cases} \quad d(i) = \begin{cases} \text{dwN} + 1, & \text{if } i < \text{r}, \\ \text{dwN}, & \text{if } i \geq \text{r}, \end{cases}$$

$$\text{upN} = \lceil m_{\text{ft}}/m_{\text{pt}} \rceil, \quad \text{dwN} = \lfloor m_{\text{pt}}/m_{\text{ft}} \rfloor, \quad \text{r} = m_{\text{pt}} \bmod m_{\text{ft}}.$$

$\mathbf{Pr}^i$ represents the $i$-th Graph-Net block in the Processor, and MEAN$\{\cdot\}$ represents the averaging operation. We can define the mapping function for the GUnet $g_2(\mathbf{GU}\text{pt})$ in a similar fashion. We leave the detailed discussion to Appendix B.2. An example of the Uniform mapping for both Processor and GUnet is shown in Figure 2b.

**2) First-N Mapping:** The second method for parameter sharing between pre-trained and fine-tuned models is the First-N method. For Processor, the goal of this mapping method is to selectively share weights for the first set of Graph-Net blocks that are common between the pre-trained and fine-tuned models. Formally,

$$\mathbf{Pr}_{\text{ft}}^i = g_1^{\text{First-N}}(i; \mathbf{Pr}_{\text{pt}}) = \begin{cases} \mathbf{Pr}_{\text{pt}}^i, & \text{if } i \le m_{\text{pt}} \\ \text{Randomly Initialized}, & \text{if } i > m_{\text{pt}}. \end{cases}$$

We can also define the First-N mapping from a pre-trained GUnet $\mathbf{GU}_{\text{pt}}$ to a fine-tuned GUnet $\mathbf{GU}_{\text{ft}}$. An example of the First-N mapping of both pooling and message passing is shown in Figure 2c.

Regardless of the mapping methods, the stages from the GUnet must first be mapped from the pre-trained model to the fine-tuned model, then within each GUNet stage the GNBs for the Processor must be mapped to the fine-tuned model as well. In this way there is a hierarchical approach to the shared parameters.

In summary, the key difference between the Uniform and First-N mapping functions lies in the extent to which parameters from the GUnet modules are used by the fine-tuned model. The Uniform strategy may allow for more comprehensive parameter use, whereas the First-N approach might be more selective. The effectiveness of these mapping functions will be assessed in Section 4.

### 3.4.2 Parameter Restriction

Beyond parameter sharing, we implement a parameter restriction technique to enhance the generalization capabilities of the downstream models. Following Gouk et al. Gouk et al. (2020), we calculate the Frobenius distance between the pre-trained and fine-tuned model weights to apply a regularization term that penalizes discrepancies between them. Let $\mathbf{W}_{\text{pt}}$ denote the weights of the pre-trained model, and $\mathbf{W}_{\text{ft}}$ represent the weights of the fine-tuned model. Then the Frobenius norm of the difference between these two sets of weights can be expressed as:

$$\|\mathbf{W}_{\text{pt}} - \mathbf{W}_{\text{ft}}\|_F = \sqrt{\sum_{i=1}^{m} \sum_{j=1}^{n} \left( \mathbf{W}_{\text{pt}}^{(i,j)} - \mathbf{W}_{\text{ft}}^{(i,j)} \right)^2}.$$

To incorporate Frobenius distance into the training process, the regularization term is added to the loss function. The regularized loss function $\mathcal{L}_{reg}$ can be written as:

$$L_{\text{reg}} = L_{\text{task}} + \lambda \|\mathbf{W}_{\text{pt}} - \mathbf{W}_{\text{ft}}\|_F^2,$$

where $\lambda$ is a hyperparameter that controls the strength of the regularization term, and $\mathcal{L}_{task}$ represents the original task-related loss.

## 4 Experiments

We present an evaluation of our proposed pre-training and fine-tuning framework for mesh-based simulations. We begin by detailing the datasets used in our experiments, highlighting both the generalized dataset constructed for pre-training and the benchmark datasets employed for fine-tuning. Following this, we introduce the baseline models against which our approach is compared. We then present the results of our pre-training phase and evaluate the transfer learning performance.

### 4.1 Datasets

**1) For pre-training:** Since there is currently no existing work on pre-training for mesh-based physical simulations, and popular benchmark datasets contain at best a few thousands training samples,

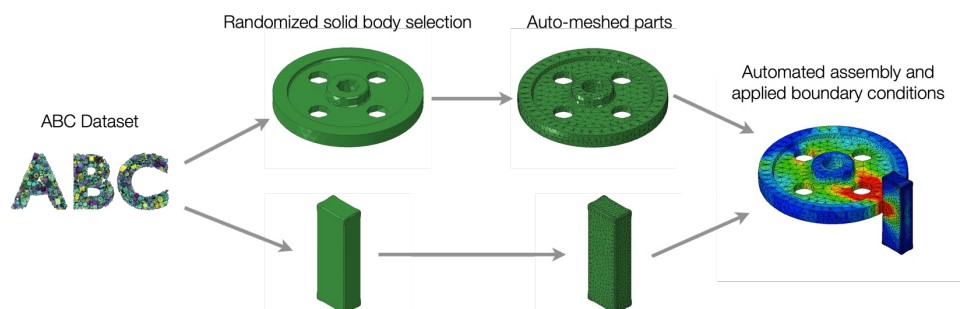

Figure 3: Randomized FEA simulation dataset using geometry from ABC dataset.

we constructed a larger and more-generalized pre-training dataset. The goal of this dataset was to have a wide variety of geometric shapes that are deformed after coming into contact with each other. We used the ABC dataset Koch et al. (2019), which is a CAD model dataset used for geometric deep learning, to get a wide sample of parts and shapes to deform. To generate a simulation in our pre-training dataset, we first randomly select two CAD geometries, then auto-mesh them with the meshing tool Shabaka Hafez & Rashid (2023). We then align the two meshed parts in 3D space and apply compressive boundary conditions to simulate the parts coming into contact. Figure 3 illustrates the workflow of the pre-training dataset construction process.

In total, we generated a pre-training dataset of 20,000 simulations by drawing pairs of geometries from a set of 400 geometry samples. Figure 4 shows several example simulations and the modes of deformation achieved through contact. Here we can see examples of mechanical contact and stress around a hole.

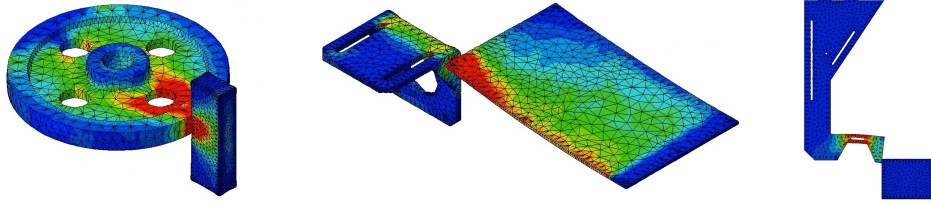

Figure 4: The FEA simulation results using ABC CAD dataset highlight various deformation modes, including compression with associated tension around a hole, as well as plate and beam bending.

**2) For Transfer Learning:** We selected two representative datasets for quasi-static mechanical simulations as benchmarks to evaluate model performance on downstream tasks: 2D Deformable Plate Linkerhägner et al. (2023) and 3D Deforming Plate Pfaff et al. (2021). These downstream task datasets represent a subspace of simulations relative to our generalized pre-training dataset, and thought to be good candidates to evaluate our fine-tuning framework. For more detailed information about the datasets, please refer to Table 2.

### 4.2 BASELINE AND METRIC

To demonstrate the generalization and effectiveness of our pre-training and fine-tuning paradigm, we used MGN Pfaff et al. (2021), a state-of-the-art model in the field of physics simulation, as the baseline for comparison. The model configurations are shown in Table 1 and more explanations can be found in Appendix C.1. We used the RMSE loss on the positions of mesh nodes $\mathcal{V}^M$ as the metric to evaluate model performance. We show the calculation of receptive field size in Appendix **??**.

### 4.3 PRE-TRAINING RESULTS

We trained both the MGN and SGUNET models on ABCD for 1 million training steps. The RMSE losses for MGN on the training and validation datasets are $8.3205 \times 10^{-4}$ and $5.8018 \times 10^{-4}$, respectively. In comparison, SGUNET achieves RMSE losses of $4.2041 \times 10^{-4}$ on the training set and $4.2657 \times 10^{-4}$ on the validation set. These results demonstrate that SGUNET outperforms MGN,

Table 1: Hyperparameter configurations for the models.

| Model | Dataset | # Message Passing (Encoder) | # Message Passing (Processor) | Mesh Receptive Field Size | # Parameters | Batch Size |
|---|---|---|---|---|---|---|
| MGN | ABCD | 5 | 13 | 18 | 1053766 | 2 |
| | Deforming Plate | 2 | 15 | 17 | 891462 | 8 |
| | Deformable Plate | 2 | 6 | 8 | 553156 | 16 |

| Model | Dataset | Pooling Ratio | # Message Passing (Encoder) | # Message Passing (GUnet) | Mesh Receptive Field Size | # Parameters | Batch Size |
|---|---|---|---|---|---|---|---|
| SGUNET | ABCD | [4, 2, 2] | 3 | 1 | 35 | 719686 | 4 |
| | Deforming Plate | [4, 2] | 4 | 2 | 29 | 894918 | 16 |
| | Deformable Plate | [2] | 2 | 2 | 9 | 569540 | 16 |

reducing the training loss by nearly 50%. Moreover, the validation loss shows that SGUNET generalizes better to unseen data while converging more effectively during training.

## 4.4 TRANSFER LEARNING PERFORMANCE

To evaluate the effectiveness of our pre-training and fine-tuning framework, we performed experiments using the ABCD dataset for pre-training. Both the MGN and SGUNET models are trained for a defined number of epochs. Subsequently, we fine-tuned these models on downstream tasks. For the Deformable Plate dataset, the models were fine-tuned for 20k steps. For the Deforming Plate dataset, the models were fine-tuned for 500k steps. We applied two parameter sharing strategies — Uniform and First-N — when loading the checkpoint of the pre-trained model. Additionally, we reduced the size of the training dataset to investigate whether our framework can decrease reliance on large volume of data. During this process, we recorded the minimum validation loss and saved the corresponding model checkpoint, which was later used to assess performance on the test dataset. All experiments are repeated 5 times with different random seeds.

**Deformable Plate:** We reduced the training set to $\frac{1}{8}$, and $\frac{1}{16}$ of its original size. The animations in Figure 5 provide an intuitive qualitative assessment. This figure presents an example from the test dataset. From these visualizations, we can observe a clear improvement in the handling of deformations at the contact area between the ball and the plate after fine-tuning. Specifically, the ball and plate no longer overlap, the plate's deformation curve conforms more closely to the ball's contour, and the deformation in areas farther from the contact point aligns more accurately with the ground truth.

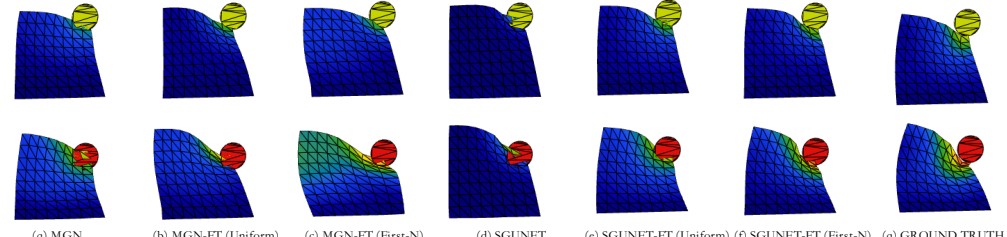

| (a) MGN | (b) MGN-FT (Uniform) | (c) MGN-FT (First-N) | (d) SGUNET | (e) SGUNET-FT (Uniform) | (f) SGUNET-FT (First-N) | (g) GROUND TRUTH |

Figure 5: Simulated meshes at various stages ($t$=30 at the top row, $t$=50 at the bottom row) for different models: MGN, MGN-FT (fine-tuned with Uniform and First-N strategies), SGUNET, SGUNET-FT (fine-tuned with Uniform and First-N strategies), and the ground truth. All models are trained on **1/8** of the original training size. The colors indicate displacement magnitude.

Figures 6 and 7 compare the roll-out validation RMSE for different models across three data scales, offering insights from various perspectives. The results demonstrate that SGUNET consistently outperforms MGN across all data scales, with lower RMSE values and faster convergence, particularly when fine-tuned. These observations are corroborated by the test dataset performance shown in Table 3. Notably, the RMSE of SGUNET fine-tuned with the Uniform strategy on $\frac{1}{16}$ of the training data is comparable to that of the model fine-tuned on the full dataset, achieving an 11.05% improvement compared to the model trained from scratch.

These results reveal that fine-tuning with the Uniform strategy further reduces the RMSE compared to the First-N strategy, demonstrating the effectiveness of this approach. Notably, SGUNET-FT with

the Uniform strategy achieves the lowest RMSE even with significantly reduced datasets, highlighting the model's robustness and efficiency in generalizing from limited data.

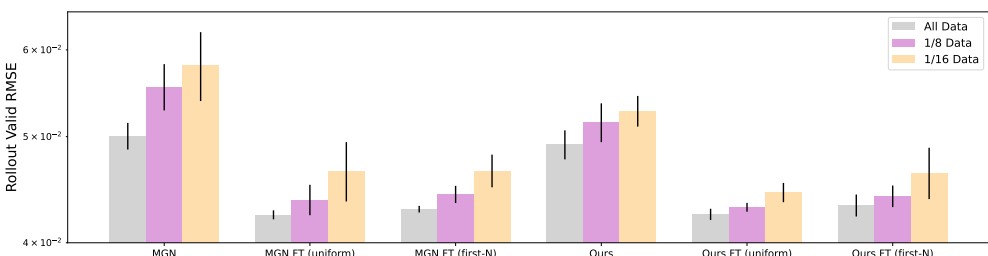

Figure 6: The best validation loss of the two models on the Deformable Plate dataset when trained from scratch and when fine-tuned.

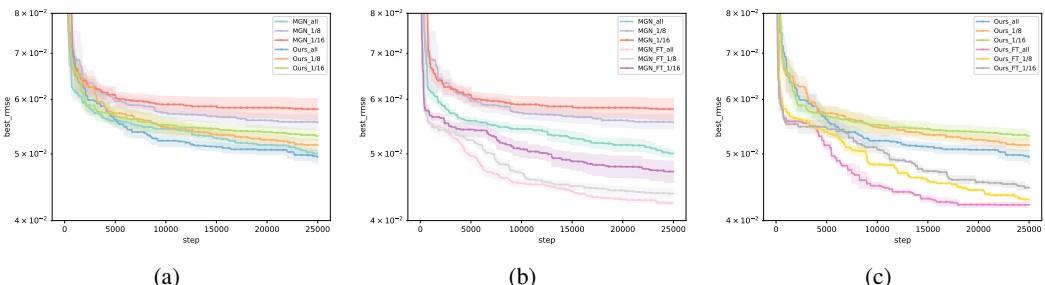

| (a) | (b) | (c) |
|---|---|---|

Figure 7: Comparison of the best validation loss tendencies on the Deformable Plate. (a) Two models both trained from scratch. (b) MGN trained from scratch and fine-tuned with the Uniform strategy. (c) SGUNET trained from scratch and fine-tuned with the Uniform strategy.

**Deforming Plate:** We reduced the training set to $\frac{1}{4}$ and $\frac{1}{8}$ of its original size. The images in Figure 8 offer a qualitative assessment by showing an example from the test dataset. The visualizations reveal that MGN performs poorly when trained from scratch as the displacement of the plate was concentrated tightly around the ball. Although fine-tuning improved MGN's predictions, the displacement area it predicts is still much smaller than the ground truth. In contrast, SGUNET, especially SGUNET-FT, delivered much more accurate predictions. This accuracy could be attributed to the larger receptive field size of SGUNET compared to MGN.

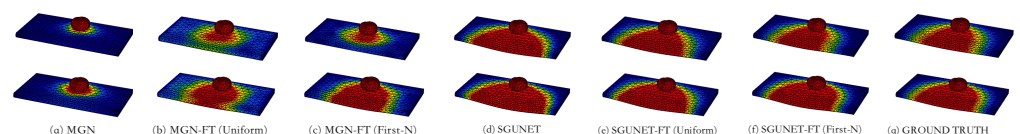

Figure 8: Simulated meshes at various stages ($t$=200 at the top row, $t$=300 at the bottom row) for different models: MGN, MGN-FT (fine-tuned with Uniform strategy), SGUNET, SGUNET-FT (fine-tuned with Uniform strategy), and the ground truth. All models are trained on **1/8** of the original training size. The colors indicate displacement magnitude.

Figures 9 and 10 compare the roll-out validation RMSE, while Table 4 presents the performance of different models on the test dataset. The results reveal that — consistent with the findings on Deformable Plate — (1) Fine-tuned models consistently outperform those trained from scratch across all dataset scales. Notably, reduction on training data does not lead to much worse performance, especially on the proposed SGUNET. (2) Fine-tuned models also exhibit faster convergence speeds. For instance, as shown in Figure 10c, SGUNET-FT on $\frac{1}{8}$ of the training data reaches the RMSE value that its counterpart requires 500k steps to achieve, in just 200k steps.

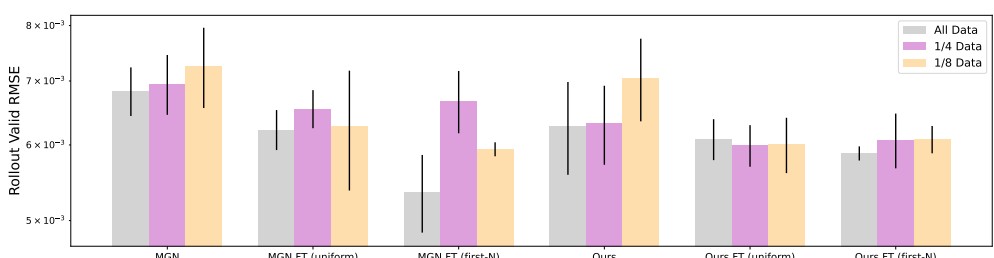

Figure 9: The best validation loss of the two models on the Deforming Plate dataset when trained from scratch and when fine-tuned.

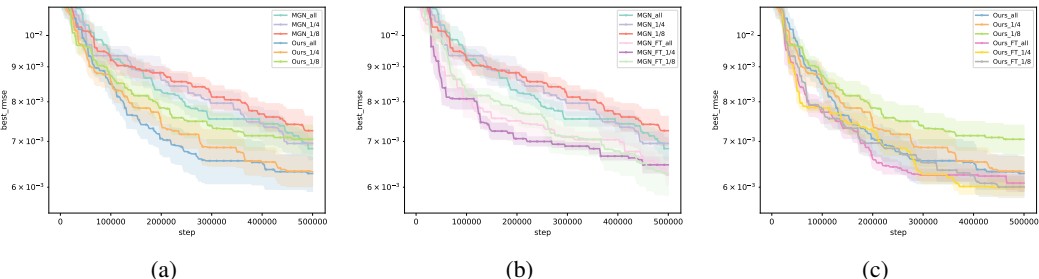

Figure 10: Comparison of the best validation loss tendencies on the Deforming Plate. (a) Two models both trained from scratch. (b) MGN trained from scratch and fine-tuned with the Uniform strategy. (c) SGUNET trained from scratch and fine-tuned with the Uniform strategy.

## 5 DISCUSSION

In this paper, we introduce a novel pre-training and fine-tuning framework tailored specifically for mesh-based physical simulations. Our approach uses a scalable graph U-net (SGUNET), which is defined in a modular and configurable manner to facilitate the parameter sharing process for transfer learning. We constructed a dataset for pre-training, i.e. ABCD, and utilized it to pre-train the models. Through extensive experiments, we demonstrate that not only does SGUNET outperform MGN, a SOTA model in this field, but also both models achieve improvements in performance across various dataset scales when fine-tuned. Notably, the fine-tuned models reduce their dependence on the training data.

Despite the promising results, there are some limitations that warrant further exploration. First, we have evaluated our framework only in the context of quasi-static simulations. Future work could extend it to a broader range of physical systems to assess its versatility and effectiveness in more dynamic scenarios. Second, our current transfer learning methods, which includes two strategies for parameter sharing and one for parameter restriction, have proven effective, exploring alternative and more advanced transfer learning techniques could offer valuable opportunities for future research.

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

## A  DATASET DETAILS

Three quasi-static datasets, ABCD, Deforming Plate, Deformable Plate are used in our experiments. All datasets are simulated as hyper-elastic deformations with linear elements. For ABCD, it includes 20,000 trajectories for pre-training. For Deforming Plate, it contains 1200 trajectories in total, we split it into 1000/100/100 for training, validation and testing. For Deformable Plate, we use a split of 675/135/135 for training, validation and testing.

In the experiments, we repeated the fine-tuning on Deformable Plate and Deforming Plate 5 times. Each time, we shuffled the data splits for training, validation and testing while maintaining the ratio.

Table 2: Basic statistics for datasets.

| pre-training | Dataset | # $|\mathcal{V}^M|$ (avg.) | # $|\mathcal{V}^E|$ (avg.) | # $|\mathcal{E}^{M,M}|$ (avg.) | # $|\mathcal{E}^{E,E}|$ (avg.) | # $|\mathcal{E}^{E,M}|$ (avg.) | # Steps | Mesh Type | Dimension |
|---|---|---|---|---|---|---|---|---|---|
|  | **ABCD** | 4445 | 12944 | 42919 | 57052 | 51777 | 20 | Tetrahedral | 3 |
| Transfer Learning | Dataset | # $|\mathcal{V}^M|$ (avg.) | # $|\mathcal{V}^E|$ (avg.) | # $|\mathcal{E}^{M,M}|$ (avg.) | # $|\mathcal{E}^{E,E}|$ (avg.) | # $|\mathcal{E}^{E,M}|$ (avg.) | # Steps | Mesh Type | Dimension |
|  | **Deforming Plate** | 1270 | 4038 | 12718 | 15648 | 16154 | 400 | Tetrahedral | 3 |
|  | **Deformable Plate** | 138 | 183 | 648 | 515 | 549 | 50 | Triangular | 2 |

## B  METHOD DETAILS

### B.1  DFS-POOLING

We provide the pseudocode for generating the cluster index vector, which is utilized for graph pooling in the GUnet module.

---

**Algorithm 1:** Get nodes clustering index

---

**Data:** adjacency matrix $\mathbf{A}$, pooling ratio $p$, material index vector $\mathbf{m}$, number of nodes $n$

**Result:** cluster index vector $\mathbf{c}$

```
 1  Function DFS (nid, mat):
 2      c.get(nid) ← cid;
 3      cnt ← cnt + 1;
 4      if cnt ≥ p then
 5          cid ← cid + 1;
 6          cnt ← 0;
 7      end
 8      for nnid ∈ [0, 1, ..., n − 1] do
 9          if nnid ∈ left ∧ m.get(nnid) == mat then
10              left.remove(nnid);
11              DFS (nnid, mat);
12          end
13      end
14  return
15  Let cid ← −1, cnt ← 0;
16  Let left ← set(0, 1, ..., n − 1), c ← [−1]_n;
17  while left is not empty do
18      nid ← left.pop(0);
19      mat ← m.get(nid);
20      if c.count(cid) > 0 then
21          cid ← cid + 1;
22          cnt ← 0;
23      end
24      DFS (nid, mat);
25  end
26  return c
```

---

## B.2 ALIGNMENT OF GUNET

A GUnet module comprise $L$ stages. To align a GUnet with $L_{\text{pt}}$ stages in the pre-trained model with a GUnet having $L_{\text{ft}}$ stages in the fine-tuned model, it is essential to construct a mapping function that build the connection between stages of the two models. Similar to the alignment of the Processor module, we design two mapping methods. Let $g_2(\mathbf{GU}\text{pt})$ denote the function that maps a GUnet $\mathbf{GU}\text{pt}$ in the pre-trained model to a GUnet $\mathbf{GU}_{\text{ft}}$ in the fine-tuned model. For **Uniform Mapping**,

$$
\mathbf{GU}_{\text{ft}}^i = g_2^{\text{uni}}(i; \mathbf{GU}_{\text{pt}}) = \begin{cases} g_1(\mathbf{GU}_{\text{pt}}^{\lfloor i/\text{upN}\rfloor}), & \text{if } L_{\text{pt}} < L_{\text{ft}} \\ \text{MEAN}_{j=st(i)}^{st(i)+d(i)}\{g_1(\mathbf{GU}_{\text{pt}}^j)\}, & \text{if } L_{\text{pt}} > L_{\text{ft}} \\ g_1(\mathbf{GU}_{\text{pt}}^i), & \text{if } L_{\text{pt}} == L_{\text{ft}}. \end{cases}
$$

For **First-N Mapping**,

$$
\mathbf{GU}_{\text{ft}}^i = g_2^{\text{First-N}}(i; \mathbf{GU}_{\text{pt}}) = \begin{cases} g_1(\mathbf{GU}_{\text{pt}}^i), & \text{if } i \leq L_{\text{pt}} \\ \text{Randomly Initialized}, & \text{if } i > L_{\text{pt}}. \end{cases}
$$

The calculations for upN, dwN, $st(i)$, and $ed(i)$ are nearly the same as those in scaling the Processor. The only difference is to replace $m_*$ with $L_*$. Here, $\mathbf{GU}^i$ denotes the Processor at the $i$-th layer of the GUnet. Prior to the alignment between GUnets, the Processor should be aligned up using the function $g_1$ to ensure consistency.

## B.3 HOW TO CALCULATE RECEPTIVE FIELD SIZE

The receptive field size is defined as the maximum distance from which the central node can aggregate information from other nodes. Let the receptive field in the $i$-th stage be denoted by $r_i$, the pooling ratio by $p_i$, and the number of message passing steps in the GUnet's Processors by $m^{\text{GU}}$. Due to the presence of a Processor at the bottom of the GUnet, we have $r_L = m^{\text{GU}}$. Given $r_i$, the receptive field for the $(i-1)$-th stage can be calculated as $r_{i-1} = (r_i + 1) \cdot p_i - 1$. By applying this recursively, the receptive field of the central node in the first stage is determined as $r_0 = (m^{\text{GU}} + 1) \cdot \left(\prod_{i=0}^{L-1} p_i + 1\right) - 2$. Prior to reaching the first stage of the GUnet, the graph has already undergone $m^{\text{Enc}}$ steps of information aggregation through the Processor in the Encoder. Therefore, the overall receptive field size is $r = m^{\text{Enc}} + (m^{\text{GU}} + 1) \cdot \left(\prod_{i=0}^{L-1} p_i + 1\right) - 2$.

## B.4 MORE INFORMATION ABOUT HETEROGENEOUS GRAPH

As described in Section 3.2, each node and edge type has a distinct feature matrix. Specifically, the feature at time $t$ is constructed as follows: 1) for $v_i \in \mathcal{V}^M$, the feature is given by $n_i \big|\big| (\mathbf{x}_i^t - \mathbf{x}_i^0)$; 2) for $v_i \in \mathcal{V}^E$, it is represented as $\lambda_i \big|\big| \mu_i \big|\big| (\mathbf{x}_i^t - \mathbf{x}_i^0)$; 3) for $e_{i,j} \in \mathcal{E}^{\text{src,tgt}}$, the feature is $\mathbf{x}_{ij}^0 \big|\big| |\mathbf{x}_{ij}^0| \big|\big| \mathbf{x}_{ij}^t \big|\big| |\mathbf{x}_{ij}^t|$. Here, $n_i$ is a binary indicator (0 or 1) representing whether the mesh node $v_i$ is a normal or boundary node, $\lambda_i$ and $\mu_i$ are mechanical properties of the material, $\mathbf{x}_i^t$ denotes the world coordinates of node $i$ at time $t$, and $\mathbf{x}_{ij}^t$ refers to the relative world position between nodes $i$ and $j$ at time $t$. The operator $\big|\big|$ denotes concatenation, while $|\cdot|$ refers to the $L_2$ norm.

Following the approach in previous work Pfaff et al. (2021), we construct edges between the plate and the ball in the Deforming Plate and Deformable Plate datasets based on the distance between endpoints. Specifically, the distances used for ABCD, Deforming Plate, and Deformable Plate are 0.0003, 0.003, and 0.05, respectively. Unlike MGN, which connects mesh nodes directly, we establish edges between element nodes.

Previous work transforms the mesh data from the Deforming Plate and Deformable Plate datasets into homogeneous graphs, where mesh vertices are represented as graph nodes. As a result, these datasets only capture the physical information of the mesh vertices and lack material properties. To address these limitations, we 1) use the average position of the mesh nodes to represent the position of the corresponding element node, and 2) set the material properties $\lambda$ and $\mu$ to zero.

## C  EXPERIMENT DETAILS

### C.1  MODEL CONFIGURATION

**MGN:** For the Deforming Plate and Deformable Plate datasets, we adhere to the settings outlined in the original paper Pfaff et al. (2021); Linkerhägner et al. (2023). For the ABCD dataset, which involves larger-scale mesh sizes, more message passing steps are required. Since increased message passing steps lead to higher memory consumption and longer training times, we balance effectiveness and efficiency by setting the message passing steps in the Encoder and Processor to 5 and 13, respectively.

**SGUNET:** We configure our model based on two key objectives: 1) achieving a larger mesh receptive field size and 2) maintaining a model size that is comparable to or smaller than that of MGN. We evaluate the model performance across several configurations and select the one that performs best.

**Noise Std.:** As we adopt the next-step prediction approach, adding noise to the input data is essential to enhance robustness during inference. We follow the noise settings specified in the original paper Pfaff et al. (2021); Linkerhägner et al. (2023) for the Deforming Plate and Deformable Plate datasets, which are 0.003 and 0.05, respectively. For the ABCD dataset, given that it involves fewer roll-out steps, we use a smaller noise standard deviation value of 0.0003.

### C.2  LOSS

The task-related objective across all datasets is unified as the mean squared error (MSE) loss of the normalized delta displacement between nodes over two steps. This can be expressed as:

$$\mathcal{L}_{task} = \frac{1}{|\mathcal{V}^M|+|\mathcal{V}^E|} \sum_{v \in \mathcal{V}^M \cup \mathcal{V}^E} \left\| \widetilde{\mathbf{x}}_v^{\,pred} - \widetilde{\mathbf{x}}_v^{\,GT} \right\|^2 ,$$

where $\widetilde{\mathbf{x}}_v^{\,pred}$ and $\widetilde{\mathbf{x}}_v^{\,GT}$ represent the normalized predicted and ground truth displacements of node $v$ over two steps, respectively.

### C.3  ADDITIONAL EXPERIMENTAL RESULTS

We present the experimental results of MGN and SGUNET on the Deformable Plate and Deforming Plate datasets in Table 3 and Table 4, respectively. These results are obtained by loading the checkpoint from the best-performing step on the validation set, followed by inference on the corresponding test datasets. To facilitate comparison, the best result for each training data size is highlighted in bold, while the second-best result is underlined.

Table 3: The performance of the two models on the test dataset for Deformable Plate when trained from scratch and when fine-tuned.

| Model | Method | All Data | $\frac{1}{5}$ Data | $\frac{1}{10}$ Data |
|---|---|---|---|---|
| | From scratch | 0.062391±0.0106 | 0.064114±0.0046 | 0.070436±0.0065 |
| MGN | Fine-tuned (First-N) | 0.056409±0.0052 | 0.057404±0.0015 | 0.058858±0.0052 |
| | Fine-tuned (uni) | **0.054644±0.0029** | **0.055432±0.0032** | 0.060024±0.0030 |
| | From scratch | 0.059615±0.0005 | 0.063806±0.0085 | 0.064714±0.0096 |
| SGUNET | Fine-tuned (First-N) | 0.057769±0.0058 | 0.059909±0.0062 | 0.061929±0.0060 |
| | Fine-tuned (uni) | 0.056966±0.0061 | 0.057517±0.0044 | **0.057560±0.0034** |

Table 4: The performance of the two models on the Deforming Plate dataset when trained from scratch and when fine-tuned.

| Model | Method | All Data | $\frac{1}{5}$ Data | $\frac{1}{10}$ Data |
|---|---|---|---|---|
| | From scratch | 0.007058±0.0009 | 0.007068±0.0006 | 0.007477±0.0008 |
| MGN | Fine-tuned (First-N) | **0.005903±0.0008** | 0.006977±0.0008 | 0.006350±0.0005 |
| | Fine-tuned (uni) | 0.006363±0.0006 | 0.006523±0.0007 | 0.006535±0.0009 |
| | From scratch | 0.006402±0.0008 | 0.006585±0.0007 | 0.007045±0.0008 |
| SGUNET | Fine-tuned (First-N) | 0.006071±0.0002 | **0.005993±0.0003** | **0.006006±0.0004** |
| | Fine-tuned (uni) | 0.006173±0.0005 | 0.006140±0.0005 | 0.006272±0.0006 |

Figure 11 and Figure 12 provide supplementary animations, where the models are trained on the full training dataset.

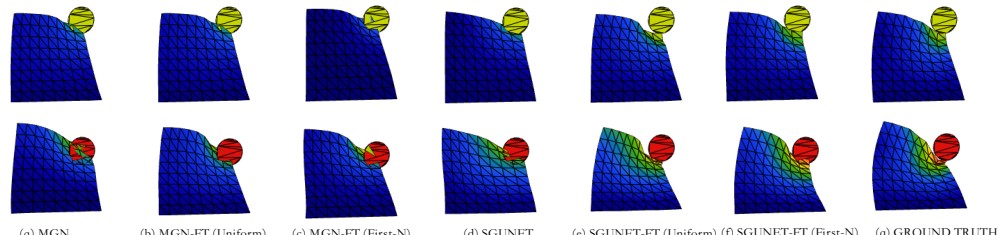

(a) MGN    (b) MGN-FT (Uniform)    (c) MGN-FT (First-N)    (d) SGUNET    (e) SGUNET-FT (Uniform)   (f) SGUNET-FT (First-N)    (g) GROUND TRUTH

Figure 11: Simulated meshes at various stages ($t$=30 at the top row, $t$=50 at the bottom row) for different models. All models are trained on the **full** training dataset. The colors indicate displacement magnitude.

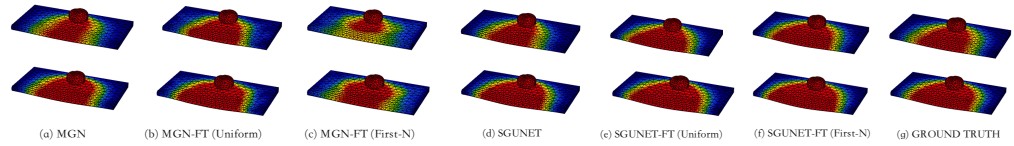

(a) MGN    (b) MGN-FT (Uniform)    (c) MGN-FT (First-N)    (d) SGUNET    (e) SGUNET-FT (Uniform)   (f) SGUNET-FT (First-N)    (g) GROUND TRUTH

Figure 12: Simulated meshes at various stages ($t$=200 at the top row, $t$=300 at the bottom row) for different models. All models are trained on the **full** training dataset. The colors indicate displacement magnitude.

