# OpenReview forum: "Transfer learning in Scalable Graph Neural Network for Improved Physical Simulation"
_ICLR.cc/2025/Conference — ICLR 2025 Conference Withdrawn Submission_

### Official Review · Reviewer_MR2f · 2024-10-23

**Soundness:** 3
**Presentation:** 3
**Contribution:** 3
**Rating:** 6
**Confidence:** 2

**Summary:**

The paper introduces Scalable Graph U-Net, a novel model for improving physical simulations using Graph Neural Networks. The paper introduces mapping functions to align parameters between pre-trained and fine-tuned SGUNET models, enabling effective transfer learning despite differences in model configurations. A new dataset named ABC Deformable with 20,000 simulations is created to pre-train the model.

**Strengths:**

1. The introduction of Scalable Graph U-Net with Depth First Search pooling is a creative and original contribution. This approach efficiently handles varying mesh sizes and resolutions in physical simulations, addressing a practical limitation in current GNN-based simulators.
2. The application of transfer learning to GNNs for physical simulations is a relatively unexplored area. The paper innovates by proposing mapping functions to align parameters between pre-trained and target models, making it easier to transfer learned knowledge between different configurations of SGUNET.
3. The architecture of SGUNET and the experiment settings are clearly presented.

**Weaknesses:**

I am not familiar with this field.
1.  While the paper claims that SGUNET can handle different mesh sizes and resolutions, the experiments focus primarily on 2D and 3D quasi-static mechanical simulations. The authors can include experiments on more diverse and dynamic tasks if possisible, such as fluid dynamics, thermal simulations, or multi-physics interactions. This would better demonstrate the generalizability of the transfer learning approach across various physical domains.
2. Although the paper claims improvements in training efficiency with reduced datasets, it does not provide detailed information on resource consumption (e.g., GPU memory, training time). A comparative analysis of training efficiency under different data scales would better showcase the model’s practical advantages.

**Questions:**

See above.

---

### Official Review · Reviewer_Fc4q · 2024-10-25

**Soundness:** 2
**Presentation:** 3
**Contribution:** 2
**Rating:** 3
**Confidence:** 5

**Summary:**

This paper introduces a new multigrid method to coarsen meshes and process them at different scales with a Graph Neural Network. This coarsening is based on spatial pooling. The authors also introduce a new transfer learning task by creating a large datasets of pairs of 3D solids. They demonstrate that GNNs can benefit from transfer learning, similarly to CNNs or LLMs, on 2 usual benchmarks of for deformable solid.

**Strengths:**

- a new multigrid method based on spatial pooling. This allows the model the process meshes at different scale and improve the speed of information propagation.
- this is to the best of my knowledge the first real analysis of transfer learning capacities for mesh-based graph neural networks.
- the novel dataset is interesting and offers a new powerful strategy to pre-train graph neural networks for solid deformation.

**Weaknesses:**

- the method is only compared to Mesh Graph Net (MGN). Since then, multiple improvements have been made, mostly based on multi grid architecture. The paper lacks a proper comparison with those methods. For example but not limited to: [1], [2] and [3].
- No ablation is performed regarding the new methodology (pooling ratio, number of layers, etc).
- Results and Table 3 seems to indicate that given enough data points, after transfer learning, the SguNet method performs worse than MGN. This and the lack of comparison with other Multi grid method does not really prove that this method should be a new standard.
- When comparing the fine-tuned models and the one trained from scratch, the total number of training steps is not taken into account. This leads to false comparison in my opinion.

**Questions:**

- l51: Previous multi-grid methods do offer dynamic scaling to counterbalance this. Was it tried?
- l53: Did you try to keep the same architecture and simply apply a trained model to another dataset? I am not sure those considerations are as "blocking" as they are presented to be.
- l67: Did you try to pretrain on 2D and finetune on 3D? This could be achieved with some sort of feature padding by adding $z=0$ coordinates and features in the 2D dataset, for example. This could also allow you to train a good model for 2D and reuse it for different asks.
- l67: Did you try to pretrain on the Deforming Plate Dataset before finetuning on the ABC dataset?
- Figure 2: You use a $concatenation$ operation as a skip connection. Why not use a $\sum$ as it is usually done?
- l242/245: I don't see how a bigger perform would perform worse. Thus, why not simply pretrain a large model before finetuning it directly? Unless you can show that in some cases, having too many message passing steps will lower performances but I don't know such case.
- l254: Did you try keeping the same encoder and decoder?
- Table 1: Given the number of parameters, you seem to be using a much smaller hidden size than the usual 128. Why? Since it was shown that 128 is a good plateau value for the Deforming Plate dataset, one might question the relevance of the results while the models are not optimized.
- Table 6: What performances do you obtain when training a model from scratch for the same amount of steps as the finetuned one (1M+ steps)?
- Figure 6 and Figure 9: Given the large range of standard deviation, can you perform $t$-tests to compare the results more precisely?

**Changes**

- l145: Wrong $ref$ for the appendix.
- l154: Acronym $MLP$ was never explained
- l371: Wrong $ref$ for the appendix.

**Additional comments**

- l70: At the moment, I am a bit skeptical of those performance comparison since you do not make any comparison with any recent architecture.
- l79-85: What about other unsupervised training method such as Physics Informed models?
- l46/47: regarding transfer learning, one could argue it is much older than the papers mentioned.
- l263 to 271: I find the formulations very hard to read.

[1] : LEARNING FLEXIBLE BODY COLLISION DYNAMICS WITH HIERARCHICAL CONTACT MESH TRANSFORMER

[2] : Simulating Continuum Mechanics with Multi-Scale Graph Neural Networks

[3] : Efficient Learning of Mesh-Based Physical Simulation with BSMS-GNN

---

### Official Review · Reviewer_xy1X · 2024-10-29

**Soundness:** 2
**Presentation:** 3
**Contribution:** 2
**Rating:** 5
**Confidence:** 4

**Summary:**

This paper introduces a transfer learning strategy for transferring pretrained GNN to other related physical simulation tasks. The main technique contributions include the Scalable Graph UNet, the parameter weight sharing, and restriction schemes. The proposed method achieves good results compared to baselines on the transfer learning tasks in experiments.

**Strengths:**

1. The structure of this paper is well organized.
2. The newly defined transfer learning tasks for physical simulation can be interesting and novel.

**Weaknesses:**

1. The literature review in this article is not comprehensive and includes many older studies. Only discussing the mesh-based simulation is not convincing, the GNN-based physical simulation and its related development should also be contained. I suggest dividing the related work section into two parts: the first part on physics simulation based on GNN, and the second part on mesh-based simulation.

I have listed several recent references and hope the author can discuss and compare them in the paper.

GNN-Based Simulator:
”Equivariant graph neural operator for modeling 3d dynamics“ 2024
"DEL: Discrete Element Learner for Learning 3D Particle Dynamics with Neural Rendering." 2024.
“A Neural Material Point Method for Particle-based Simulations” 2024

Mesh-Based simulation:
Magnet: A graph u-net architecture for mesh-based simulations. 2024. By the way, this paper uses a similar architecture to the reviewed paper.

2. Line 145 : Appendix ?? should be a reference error.

3. As for the parameter restriction, The author says that it is necessary to calculate the Frobenius norm of the difference between Wp and Wt (sec. 3.4.2) but they often have unequal shapes. How should the difference be calculated?

4. It would be best to label the equations with numbers.

5. Are "the number of stages" and "message passing steps" parameters that need to be manually specified, or can they be adaptively adjusted? I believe adaptive adjustment could enhance the generalizability of this method. Because these two parameters are highly close to the data size and simulation settings.

6. As a fine-tuning method, I strongly recommend comparing the time required for fine-tuning and the model size, as this can highlight the method's contributions to fine-tuning efficiency. The author seems to have only compared the effect of the amount of data required for fine-tuning the final results. As I said before, the model size and time required for finetuning are also important for practicability. Additionally, MGN is a relatively old baseline. If possible, I hope the author can include more recent studies for comparison.

**Questions:**

See weakness.

---

### Official Review · Reviewer_QqvV · 2024-11-05

**Soundness:** 2
**Presentation:** 3
**Contribution:** 2
**Rating:** 5
**Confidence:** 4

**Summary:**

This paper mainly makes two contributions to graph-based physical simulations. The first contribution is using the depth-first-search random walk strategy to improve the graph pooling module. The second contribution is aligning pre-trained model weights with fine-tuned model weights and proposing two strategies for weight alignments. Experimental results demonstrate some empirical success of the proposed method.

**Strengths:**

(1) The presentation of the major idea is decent. This reviewer can understand the core idea of the proposed method;

(2) Improving the scalability of GNNs for physical dynamics simulations is crucial. This reviewer acknowledges the importance of the addressed problem.

**Weaknesses:**

(1) The technical contribution of this paper appears somewhat limited. While enhancing the graph pooling module and model weights could be beneficial, the techniques employed are not particularly innovative. Both the graph pooling strategy and the model weight alignment strategy have been previously proposed. Applying these strategies to the scenarios mentioned in the work does not present any novel challenges.

(2) The motivation behind the study is not entirely clear. As a reviewer, I am still uncertain as to why the current graph U-Net is not scalable enough, how the use of DFS can help GNN adapt to different mesh sizes, and why existing solutions cannot achieve this objective.

(3) The presentation lacks a systematic and unified approach. At present, it appears that two strategies are proposed to enhance overall system performance in separate sections, but these strategies do not seem closely connected.

(4) While I am not entirely familiar with the general physical dynamics simulation research works, the experiments conducted in this study still seem insufficient. Currently, the experimental evaluation lacks a systematic benchmark to robustly support the superiority of the proposed method, which undermines the significance of the proposed method.

**Questions:**

Why do we need pre-training and fine-tuning paradigms for the physical dynamics simulation? And also why DFS can help adapt to different resolutions of meshes . The motivation is still not very clear. Could the authors further elaborate on the motivation?

---

### Note · Authors · 2024-11-21

**Comment:**

We appreciate the detailed reviews, which provide great insights and constructive advice for the paper's contents.
The authors would like to spend more time for extra experiments (comparing to more SOTA models, carrying out ablation study for mentioned hyper-parameters etc.) to improve and refine the contents.

Due to limited time and resources, we were unable to fully address all reviewers' questions with solid evidence at this stage. Therefore, we have decided to withdraw our submission to allow more time for these improvements. We are grateful for the thoughtful feedback and look forward to refining the paper for future submission. Thank you again for your time and consideration.

**Withdrawal Confirmation:**

I have read and agree with the venue's withdrawal policy on behalf of myself and my co-authors.